



Comment on "Ice content and interannual water storage changes of an active rock glacier in
the dry Andes of Argentina" by Halla et al. (2021).

W. Brian Whalley
Department of Geography, University of Sheffield, Sheffield UK
*Correspondence to:* W. Brian Whalley (b.whalley@sheffield.ac.uk)

**Abstract.** Recently published work on water preservation in Chile assume that 'permafrost'
(cryogenic) rock glaciers are dominant. Melt pond development shows that rock glaciers are
glacier-derived ('glacigenic') rather than of permafrost origin.

Halla et al. (2020) make a useful contribution to estimating the water content of rock glaciers at 'Dos
Lenguas' in Chile (69º 47' 12" W, 30º 14' 48" S). However, their interpretation (Figure 10) relies on the
assumption that it is 'talus rock glacier', a body of creeping permafrost unrelated to any glacier.
Although commonly held, this origin is not supported by rheology (Whalley and Azizi, 1994). Further,
the Dos Lenguas (DL) site shows no rock glacier formation in or from the extensive local talus. The
glacier ice core ('glacigenic') model better explains formation and flow (Whalley and Azizi, 2003).
Gruben rock glacier, taken to be a 'typical' permafrost-derived rock glacier, is actually of Little Ice Age
origin and is glacier-ice cored (Whalley, 2020). At DL, a small glacier formed in a south-facing hollow
then covered by insulating weathered rock debris. To the west (6.5 km) of DL there are several rock
glaciers where glacier ice could collect and be buried. The largest of these (Figure 1) lies below a glacier
and debris-covered glacier. Over the last 15+ years glacier melting has produced substantial surface
pools. Some 16 km (30° 09' 21"S, 69°54′ 40"W) from DL, the Tapado-Las Talas glacier-rock glacier
complex has similar features. Monnier et al. (2014) show a debris-covered glacier with melt
(thermokarst) pools merging with a rock glacier, itself over-riding a moraine sequence. Schaffer et al.
(2019) considered this a complete rock glacier sequence (Tg) below the Tapado glacier with the debris-
covered section being 'glacigenic' (their Figure 3). The neighbouring Las Tolas rock glacier (Tc) was
viewed as 'cryogenic' (permafrost-periglacial). There is no visible glacier component in the cirque above
Tc although Google Earth images (2017) show copious snow collection and crevasse features (noted
by Schaffer et al.) on the steepest section. As with the rock glaciers west of DL, the simplest explanation
for all these features is glacigenic. The seismic traces used by Schaffer et al. to differentiate between
Tc and Tg are probably due to the complex relationships of ice-snow and debris supply. The geophysical
data supplied by Milana and Güell (2008) and Halla et al. (2020) will be useful in the interpretation of
these factors in glacier/rock glacier formation and the development of models to estimate water storage
potential.



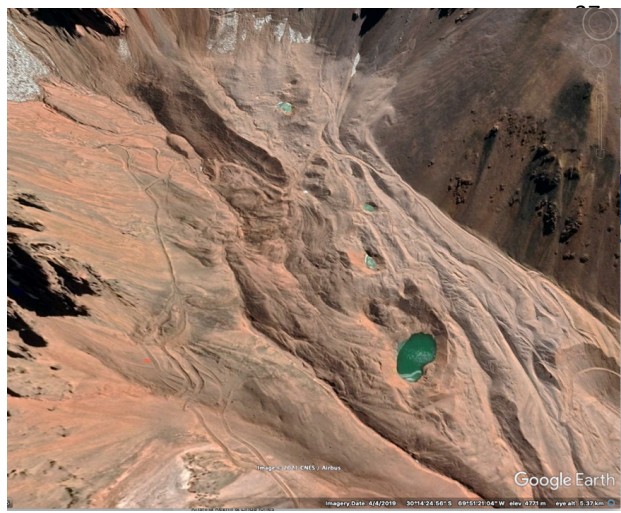

Figure 1. Glacigenic rock glacier located at 30º 14' 29" S, 69º 51' 15" W. © Google Earth/CNES/Airbus. Melt pools show ablation of massive, glacier-derived, ice under a debris cover. A permafrost (talus-derived) feature would show 'isovolumetric' melting of ice in pore spaces and thus have rather different water storage capability from a glacier core.

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
