# Peer review of "Comment on "Ice content and interannual water storage changes of an active rock"

_The Cryosphere, 2021_

## Author Response (AR1)

**Comment on "Ice content and interannual water storage changes of an active rock glacier in the dry Andes of Argentina" by Halla et al. (2021).**

W. Brian Whalley

Response 3 January 2023

I have added a final summary paragraph as requested.
Updated the reference to Halla et al. 2021
Used a decimal latitude longitude rather than a d,m,s

---

## Author Response (AR2)

**Comment on "Ice content and interannual water storage changes of an active rock glacier in the dry Andes of Argentina" by Halla et al. (2021).**

W. Brian Whalley

Response 17 January 2023

Dear Regula,

Thank you for the editing. The changes are show in yellow highlight on the associated 'changes' version.

I take your point about the Tc and Tg text references, although they do not refer to Fig 1. I had used these in my efforts to keep things brief and use the labels of some of the authors referred to. I have now deleted these (Tc and Tg) and put brief explanations in their place. These new additions are in green. I have also used the decimal latitude-longitude (dLL) notation for much more precise location of the features referred to. I had already done this with the first citation of location. This lengthens the comment slightly but is much less ambiguous. Furthermore, using dLL makes it easier for the reader to see (via Google Earth) what is meant as well as locate places and features by other authors. These additions do not change my expressed views in any way, it is only for increased clarity.

I hope these changes are acceptable.

Incidentally, since submitting my original note, I have been working on this [dLL] notation and its use in several articles (which I can send if you wish) and it seems to be getting accepted. It has great value in making clear the general location where there may only be a large scale map, perhaps with latitude and longitude (as d/m/s) round the edge, the original Halla et al paper is once such). It can take be difficult to locate small features and where only local feature names are given.

---

## Author Response (AR3)

Comments and responses to referees and authors.
My response:
Thank you for your remarks.
1. My commentary is based on observations via Google Earth imagery. This makes it
possible for any reader to look at the field evidence and surrounding areas. Charles
Darwin noted, 'How odd it is that anyone should not see that observation must be for or
against some view if it is to be of any service' (Ayala, 2009). This quotation highlights
issues in the philosophy of science and the nature of evidence both of which I touch upon
in my responses hereafter. I have numbered the main points sequentially for the benefit
of the reader.
2. My original comments, and indeed my responses posed here, are intended to show
readers the field evidence as I see it; 'it is essential to the scientific process that any
hypothesis be ''tested'' by reference to the natural world that we experience with our
senses' (Ayala, 2009).
3. Although it may not be 'possible to determine the origin of rock glaciers', the reviewer
acknowledges that my argument is 'sufficiently convincing' to warrant using the
glacigenic model for the Dos Lenguas (DL) rock glacier. My comments are based on
observations from various glacier-rock glacier landsystems in the in the area. I chose to
illustrate it with one specific example, but I fill in some more detail in my responses to
others below.
4. In the responses I use the following convention to help readers identify locations on
Google Earth (GE) by pasting in the numbers in the GE search bar between square
parentheses. Thus, Dos Lenguas (DL) can be identified as decimal latitude and longitude
[-30.24664,-69.78667]. A transect along the 'fall line' on the feature starts at the top with
the last term (260) being a bearing in degrees from the preceding couplet as origin: {-
30.24235,-69.76730,260}. This decimal degree convention is more useful to
georeference features at various scales and transects for recording purposes than the
traditional ° ' ". See Whalley (2021a, 2021b; collated references are at the end) for
illustrations about the notation for studying rock glaciers elsewhere.
Thank you for your comments. I fill in some detail here in direct response to your remarks
(other information is provided below). I have tried to keep these succinct and directly
related to what is particularly pertinent.
5. As my comments were primarily about field observations (see 1, supra), I only included
two papers about the rheology of ice rock mixtures. It is the mechanical nature of the mixture
model (rock/ice-snow/water/air) that determines the rheology. A thin glacier (<30m thick,
slope angle ca 10°, with an ablation-reducing debris cover) will flow at rock glacier
velocities, < 1 ma$^{-1}$. However, talus (scree or rockfill) as an 'ice sparse' composite will not
flow unless the ice content is high (perhaps >60%) and in thick ($\approx$ 20 m) deformable bands or lenses. The geophysical signature of a rock glacier at any location depends upon the field-
mixture model, as well as the volume examined, given its inhomogeneity and anisotropy. The
permafrost model correlates geophysical signatures to a formational mode for all rock
glaciers (i.e. exclusively of non-glacier origin, see 17 below). My commentary suggests there
is directly observable field evidence for a glacial origin for the deforming ice at DL. But, as
Lliboutry noted (1990) of a comment by Haeberli (1989), 'I do not deny that many (not all)
rock glaciers are below melting point at depth'.

6. Why don't all the slopes in the area show flow-features when, in a known permafrost area,
there are plentiful scree slopes? The answer is that they will do so only if there is a thick
enough body of ice, as a glacier in a conventional sense or with a thick snow/ice body
covered with debris (5). On cliffed slopes with snow avalanching, this can be achieved if
perennial snow accumulates (and is buried, perhaps sequentially, under debris). This is the
point made by reference to rheology in Whalley and Azizi (2003) and the mixture model (see
5). It is the creep of massive ice, not rock debris – even if this is in a permafrost area.
Permafrost is not necessary, but it is sufficient to keep creep rates lower than at ice pressure
melting point. As an illustration, the transect, 1: {-30.2423,-69.7670,260} down the centre of
DL rock glacier can be compared with a parallel transect, 2:{-30.24908,-69.76338,270}. The
latter, some 700 m to the south of 1, is representative of much of that mountainside and must
be under the same environmental conditions, temperature, snowfall and ablation, as the rock
glacier, 1. However, transect 2 shows no signs of flow. The reason must lie in the 'mixture
model', debris from the upper slopes has covered a perennial snowpack of a 'buried glacieret',
'buried debris-rich glacieret' or 'glacier enterré' (Lliboutry, 1961; Lliboutry, 1990). That there
is no glacier/glacieret remnant showing at 1 is because the thick ice mass necessary for flow
is covered with debris from above. The top of this original, small and confined, glacier would
have been under the cliffs in the vicinity of Google Earth locality [-30.2429,-69.7747] and
fed down gullies higher on the slope. Extant equivalents can be seen at the top of gullied
south-facing slopes in the vicinity of [-30.23512,-69.83599]. The glacier and its protecting
debris load have now crept downhill and formed the DL rock glacier. A short transect {-
30.24318,-69.77858,160} for about 150 m, i.e. some 250 m east of the Halla et al. 'root zone'
transect, is lower in the centre (by 5-10m) from the edges. This shows that ice had flowed out
of this area and has not been replaced. This effect is similar to other rock glaciers with
extending flow regimes (Whalley and Palmer, 1998, Whalley, 2021b).

7. Observations using GE brings to light further changes in surface topography of rock
glaciers, notably the appearance of pools that show melting of ice below the surface debris.
Recent coverage by GE shows meltwater pool exposures are becoming increasingly common.
Ridges and furrows, piled up in lower (snout) regions are the result of basically compressive
glacier flow with debris loads becoming increasingly thick near and at the snouts. This
inhibits melting further from upstream amounts (where the debris load is thinner). Glaciers
and rock glaciers may exhibit extending flow where, usually on steeper slopes and perhaps
more restricted valley sections, transverse ridges and furrows are replaced by irregular or
longitudinal features. Meltwater pools can form variously in them according to local
topography and thickness of the debris cover.

8. These meltwater pools can be of considerable size, that shown in my Fig 1 at [-30.2413,-
69.8542] has a water area of about 3 000 $m^2$ and has been in existence at least between 2006
– 2019 (from GE imagery). The total 'missing' volume of rock glacier is some 40 x $10^3$ $m^3$,
suggesting that the mixture model is predominantly of high percentage (massive) ice from a
buried glacier tongue. This is commensurate with the sides of a 'thermokarst depression'

95 shown (Figure 4) of Trombotto-Liaudat and Bottegal (2020) at Morenas Coloradas debris-
96 covered glacier [-32.9426,-69.3988] although the exact location is not given. Other long-lived
97 meltwater pools can be seen up-valley to the exposed glacier at Morenas Coloradas, further
98 examples can be seen in some of the images in Janke et al. (2015). Whether rock glaciers
99 extend back into visible debris free and debris-covered versions (as suggested in the
100 classification of Janke et al. (2015)) depends upon the relative inputs of glacier ice and
101 weathered debris over time. The Colina Mountain example (Janke et al., 2015, Fig. 21B) [-
102 34.3428,-70.0492] has a continuum of classes of debris-covered glacier/rock glacier with
103 surface forms that include meltwater pools [-34.3437,-70.0486] & [-34.3494,-70.0583] and
104 lateral erosion of pool with an exposed glacier ice cliff [-34.3571,-70.0718].

105 HAEBERLi

106 Thank you for your comments Wilfried.

107 9. Please note that I said, 'The geophysical data supplied by Milana and Güell (2008) and
108 Halla et al. (2020) will be useful in the interpretation of these factors in glacier/rock glacier
109 formation ...' In other words, evaluating the nature of the 'mixture model' that should be
110 applied to the rheology (6, supra) will be helpful in establishing the geophysical properties
111 and variability in rock glaciers. I am well aware of the range of geophysical results available
112 from rock glaciers and why they can be so variable (acknowledged by Referee 2) and noted
113 this in my original comment. This is also part of the review of the mixture models provided
114 by Whalley and Azizi (1994) and I do not propose to discuss this variability here as my point
115 was, and is, to look at visible forms and how they might inform us as to the origin of rock
116 glaciers. The rheology gives the landform and its details, not the variable geophysical
117 signature.

118 10. I am also aware of Gruben glacier/rock glacier and its ice-dammed lakes and the so-called
119 'periglacial part'. But readers should note that an interpretation of that rock glacier landsystem
120 suggests that the rock glacier *does* have a glacier ice core (Whalley, 2020). It is no different
121 from the observations of glacier ice cores in rock glaciers that have been recorded over the
122 years from many parts of the world, for example; Kesseli (1941), Potter et al. (1998) and
123 more recently Whalley (2021b). No amount of geophysical pleading can refute these
124 observations. It is for time, as more meltwater pools are exposed, and readers to evaluate. A
125 rough calculation (see 8, supra) shows that such meltwater pools are from the decay of
126 massive glacier ice – which is what was the case at Gruben (Whalley, 2020).

127 11. It is certainly true that boreholes and exposures do show the complex nature of ice and
128 debris in rock glaciers, see for example Janke et al. (2015) and Jones et al. (2019), especially
129 near rock glacier snouts. Because of the increasing surface debris loads down-valley, ice
130 exposures tend to be hidden by debris. However, some snout collapses can be seen in GE,
131 such as at Glockturmferner (Austria) [46.89846,10.65058], compared with earlier views
132 (Kerschner, 1983). Lliboutry described a section in the one of the four 'glaciers enterrés'
133 below the west face of Cerro Negro (Andes of Santiago). The exact location is unknown but
134 is in the vicinity of [-33.1484,-70.2367] (Lliboutry, 1961, Fig. 1). The section (Lliboutry,
135 1961 Fig. 4) and (Lliboutry, 1965 Fig.17.21) shows complex relationships between ice;
136 young, old bubbly and bubble free ice together with silt and pebbled bands. This is more
137 complex than the section shown by Trombotto-Liaudat and Bottegal (2020). Figure 8 of
138 Janke et al. (2015) shows section of a meltwater pool showing banding, similar to Gruben
139 rock glacier's drained lakes (Whalley, 2020). There is clearly much to be gained about the structures of glaciers as they become exposed at the snouts of rock glaciers. This will help in
matching geophysical attributes to structural glaciology and debris content.

12. Although there have been descriptions of rock glaciers since the early 20th C, the paper
by Wahrhaftig and Cox (1959) has become particularly import in discussion about these
features (Stine, 2013). Indeed, it has become the 'Urtext' for those believing the 'permafrost'
origin of rock glaciers promoted by Wahrhaftig and Cox. The book by Barsch (1996)
provides the stated dogma of the permafrost viewpoint. This text is followed by Barsch
(1987) who denigrates many observations of glacier ice cores. Subsequently, sins of omission
have followed by disregarding any other possibilities than the permafrost dogma, e.g. Swift et
al. (2021). Please see Whalley (2021a) where some of these wrongs are addressed.

13. Professor Haeberli, as a true believer in the Urtext and permafrost dogma, has always
maintained that rock glaciers cannot have glacier ice cores (i.e. be glacigenic). For him, this
means that not only do glacier ice cores not exist but that any continuum or equifinality does
not occur (pace Referee 2). Yet there are many reports of glacier ice in rock glaciers, as well
as the well-established work of Potter at Galena Creek that cannot be denied (although I leave
it to readers to adjudicate). Quoting many references that support a permafrost viewpoint
amounts to 'affirming the consequent' (modus tollens). In terms of swans and rock glaciers,
all swans are not white and at least some rock glacier swans are black and contain glacier ice
cores. Thus, supposition and following a particular point of view is insufficient to replace
valid contra-observations. In a Popperian sense therefore we might have to wait for contra-
indications of permafrost, or affirmation of the appearance of glacier ice by meltwater ponds.

14. I have mentioned the work of the late Professor Louis Lliboutry in reporting 'glacier
enterré' and in particular the complexities of snout stratigraphy. He also said (Lliboutry,
1990); 'I do not wish to enter into a public controversy with W. Haeberli about the origin of
rock glaciers; he has always been deaf to my arguments. Nevertheless, the readers of his
passionate assertions (Haeberli, 1989) must be aware that he intentionally omits to quote my
detailed observations in the dry Andes (Lliboutry, 1955, 1965, 1986).' Further, 'Nevertheless,
for the advancement of science, the essential point is not "must rock glaciers be left to
scientists claiming to be permafrost specialists" but "what can we learn from the existence of
rock glaciers in a given area"? I maintain that the geographical study of rock glaciers as an
extreme case of glacier fluctuations, as an indicator of favourable mass balances in the past,
or of past surges, would be much more rewarding than to consider them as a mere case of
standard permafrost, or of creeping regolith.' (Lliboutry, 1990).

Halla et al

Dear Authors. Thank you for your comments

Regarding your first point, I appreciate that your detailed work refers to a single feature. By
implication however, your findings refer to the general study of water storage in glaciers and
rock glaciers. Thus, your study becomes a part of an overall appreciation of water content in
South America and needs to accommodate a variety of findings under slightly different
climatic conditions – as you are arguing for a zonal (or morphoclimatic) interpretation.

15. I appreciate your view (third point) that, 'the assessment and discussion of the origin of a
distinct rock glacier or landform should be based on on-site specific geomorphological
characteristics (form, process, and material) of the landform. Indeed, I recently (Whalley,

2021a) I suggested that it was necessary (though geomorphological mapping) 'to recognise
and link materials (M), 'processes' (P, that is mechanisms integrated over time) and visual
categorization and geometrical information (G). In principle, this information, i.e. site
metadata, can be collected and a database interrogated to maximise geomorphological
knowledge'. I suggest above (points 6 and 9) that it is the rheological (dynamic) properties of
a feature and related to the materials, that account for the forms seen. In this it is necessary to
look at the connectivity of material movement downslope and the origin of both water/ice
and solids. Further, that other examples in the literature, which can be seen on Google Earth,
do show rheological properties that are consistent with a glacier ice core (for valley floor rock
glaciers) or a substantial snow/ice mass that has been buried by copious debris supplies from
above – which is the case at DL. As mentioned above (6) ice that collected in the vicinity of
[-30.2429,-69.7747] has moved downslope and now lies buried under the debris in the snout
lobes. That there are no 'glacial deposits, like moraines' as 'traces of a former glacier' is rather
easily explained; the rock glacier deposits *are* the moraines. A transect {-30.24316,-
69.77959,255} shows a distinct (right) lateral moraine of a former small debris-covered
glacier, with its main ice collection area at about [-30.2429,-69.7784]. This small glacier was
clearly overwhelmed by the ice and sediments of the ice rock glacier of DL.

16. It is arguable whether science should be conducted according to inductive or deductive
principles (see Ayala (2009) for basic discussion related to Darwin). Goudie and Viles (2010)
argue for an abductive view in the construction of ideas and models but in order to overcome
'prejudices and conditioning' the 'critical rationalist approach' of Karl Popper should be used
to 'attempt to disprove rather than verify our hypotheses' (Schumm, 1991). In other words,
and in this case, alternative viewpoints are not only acceptable but to be welcomed (12,
supra). Thus, my observations of meltwater pools in a wide variety of instances in the
literature, which show that ice melting is not 'iso-volumetric' supports a massive ice origin. A
theory should make predictions that can be tested. I suggest that meltwater pools will be seen
on DL around [-30.2479,-69.7850] in the next ten years to become like [-30.2413,-69.8542]
to which it is topographically similar and functionally related.

17. I shall not argue about your geophysical results – which was not my intention in the first
place – and referee 2 (supra) has already commented on these. However, you state that Dl
should be considered as a 'talus rock glacier'. I have no difficulty with the terminology only
that it must necessarily be 'creeping permafrost'. Some authors e. g. Evin et al. (1997) have
argued for 'hybrid models' and Monnier and Kinnard (2015) have discussed 'glacier-rock
glacier transitions' and Jones et al. (2019) present water content evidence from a variety of
rock glacier models. More investigations are clearly required.

18. With respect to 'surface texture, the geomorphological characteristics and spatial
connection of the rock glacier to the upslope are recommended proxies for visual
observations' (IPA, 2020) I have here outlined some reasons for considering the
characteristics at DL (and elsewhere) as indicative of glacier flow. However, the IPA
document presents a major misunderstanding of the nature of rock glaciers by concentrating
on kinematics rather than dynamics (rheological properties). Any flow mechanisms, i.e.
dynamics not just kinematics, needs to consider the full implications of the materials
involved. In other words, the IPA statement follows the pure Urtext (12) with not even
alternatives such as hybrid or equifinality possibilities.

19. I do not have space to argue my case about the IPA (2020) publication but rather point
out that in stating that 'rock glacier (or permafrost) creep has to be understand (sic) here as a generic term' (p. 6) and 'Rock glaciers, as landforms resulting from a permafrost creep
process, should not be confused with debris-covered glaciers'. (p. 11) it follows the
'exclusive' approach (5 supra). In particular, by assuming the dogma associated with the
permafrost Urtext (12) and by ignoring the glacial/glacigenic model for which there is good
evidence, it has engendered 'belief perseverance' in some sectors of the geoscience
community where there is also 'confirmation bias' that has not been assuaged by showing
falsifiers (black swans). That I have generated some discussion is a good thing, although I
return to my original quotation from Charles Darwin on observations. But thank you for your
paper and its valuable measurements.

Ayala, F.J.: Darwin and the scientific method. Proc. Nat. Acad. Sci., 106, Supp. 1, 10033-
10039, 2009.
Barsch, D.: The problem of ice-cored rock glacier. In: J.R. Giardono, J.F. Shroder, J.D. Vitek
(Eds.), Rock glaciers. Allen and Unwin, London, pp. 45-53, 1987.
Barsch, D.: Rockglaciers. Indicators for the present and former geoecology in high mountain
environments. Springer, Berlin, 331 pp. 1996.
Evin, M., Fabre, D. and Johnson, P.G.: Electrical resistivity measurements on the rock
glaciers of Grizzly Creek, St Elias Mountains, Yukon. Permafrost Periglac., 8(2), 179-
189, 1997.
Goudie, A. and Viles, H.: Landscapes and geomorphology: a very short introduction. OUP,
Oxford,  137 pp.2010.
Haeberli, W.: Glacier ice-cored rock glaciers in the Yukon Territory, Canada? J. Glaciol.,
35(120), 294-295, 1989.
Halla, C., Blöthe, J. H., Tapia Baldis, C., Trombotto, D., Hilbich, C., Hauck, C., and Schrott,
L.: Ice content and interannual water storage changes of an active rock glacier in the
dry Andes of Argentina, The Cryosphere Discussions, doi.org/10.5194/tc-2020-29,
2020.
IPA, International Permafrost Association:  IPA Action Group, Rock glacier inventories  and
kinematics. Available at:
https://bigweb.unifr.ch/Science/Geosciences/Geomorphology/Pub/Website/IPA/Guid
elines/V4/200507_Baseline_Concepts_Inventorying_Rock_Glaciers_V4.1.pdf. 2020.
Janke, J.R., Bellisario, A.C. and Ferrando, F.A.: Classification of debris-covered glaciers and
rock glaciers in the Andes of central Chile. Geomorphology, 241, 98-121, 2015.
Jones, D. B., Harrison, S., Anderson, K., and Whalley, W. B.: Rock glaciers and mountain
hydrology: A review, Earth-Sci. Rev., 193, 66–90,
https://doi.org/10.1016/j.earscirev.2019.04.001, 2019.
Kerschner, H.: Zeugen der Klimageschichte im Oberen Radurschltal. Alpenvereinsjahrbuch,
1982-83 DÖAV, 23-28, 1983.
Kesseli, J.E.: Rock streams in the Sierra Nevada, California. Geogr. Rev., 31(2), 203-227,
1941.
Lliboutry, L.: Origine et évolution des glaciers rocheux. R. Acad. Sci. (Paris), 240, 1913-
1915, 1955.
Lliboutry, L.: Les glaciers enterres et leur role morphologique, Symp. Helsinki Int. Ass. Sci.
Hydrol. Pub. 54, pp. 272-280, 1961.
Lliboutry, L.: Traité de Glaciologie, 2. Masson & Cie, Paris, 707 pp. 1965.
Lliboutry, L.: About the origin of rock glaciers. J. Glaciology, 36(122), 125-125, 1990.

Monnier, S. and Kinnard, C.: Internal structure and composition of a rock glacier in the
Andes (upper Choapa valley, Chile) using borehole information and ground-
penetrating radar. Ann. Glaciol., 54, 61–72, https://doi.org/10.3189/2013aog64a107,
2013.
Potter, J., N, Steig, E., Clark, D., Speece, M., Clark, G.T. and Updike, A.B.: Galena Creek
rock glacier revisited—New observations on an old controversy. Geogr. Ann. A,
80(3-4), 251-265, 1998.
Schumm, S.A.: To interpret the earth: ten ways to be wrong. Cambridge University Press,
Cambridge, p. 133, 1991.
Stine, M.: Clyde Wahrhaftig and Allan Cox (1959) Rock glaciers in the Alaska Range.
Bulletin of the Geological Society of America 70 (4): 383–436. Prog. Phys. Geog.,
37(1), 130-139, 2013.
Swift, D.A., Cook, S., Heckmann, T., Gärtner-Roer, I., Korup, O., Moore, J.: Ice and snow as
land-forming agents, Snow and Ice-Related Hazards, Risks, and Disasters. Elsevier,
pp. 165-198, 2021.
Trombotto-Liaudat, D., Bottegal, E.: Recent evolution of the active layer in the Morenas
Coloradas rock glacier, Central Andes, Mendoza, Argentina and its relation with
kinematics. Cuadernos Investigación Geográfica, 46(1), 159-185, 2020.
Wahrhaftig, C., Cox, A.: Rock glaciers in the Alaska Range. Geol. Soc. Am. Bull., 70(4),
383-436, 1959.
Whalley, W.B., Palmer, C. F.: A glacial interpretation for the origin and formation of the
Marinet Rock Glacier, Alpes Maritimes, France. Geogr. Ann., A, 80, 3/4 221-236,
1998.
Whalley, W. B.: Gruben glacier and rock glacier, Wallis, Switzerland: glacier ice exposures
and their interpretation, Geogr. Annaler: A, 102, 141-161, 2020.
Whalley, W.B.: Geomorphological information mapping of debris-covered ice landforms
using Google Earth: an example from the Pico de Posets, Spanish Pyrenees.
Geomorphology, https://doi.org/10.1016/j.geomorph.2021.107948, 2021a.
Whalley, W.B.: The Glacier – Rock Glacier Mountain Landsystem: an example from North
Iceland. Geogr. Ann., B, 2021b.
Whalley, W. B., and Azizi, F.: Rheological models of active rock glaciers: evaluation,
critique and a possible test, Permafrost Periglac., 5, 37-51, 1994.
Whalley, W. B., and Azizi, F.: Rock glaciers and protalus landforms: Analogous forms and
ice sources on Earth and Mars, J. Geophys. Res.: Planets (1991–2012), 108, 2003.
.